# Study of Biodiversity of Algae and Cyanobacteria of Mutnovsky and Gorely Volcanoes Soils (Kamchatka Peninsula) Using a Polyphasic Approach

**Rezeda Z. Allaguvatova** [1], **Arthur Yu. Nikulin** [1], **Vyacheslav Yu. Nikulin** [1], **Veronika B. Bagmet** [1] **and Lira A. Gaysina** [2,3,*]

1 Laboratory of Botany, Federal Scientific Center of the East Asia Terrestrial Biodiversity, 690022 Vladivostok, Russia; allaguvatova@yandex.ru (R.Z.A.); artyrozz@mail.ru (A.Y.N.); nikulinvyacheslav@gmail.com (V.Y.N.); chara1989@yandex.ru (V.B.B.)
2 Department of Bioecology and Biological Education, M. Akmullah Bashkir State Pedagogical University, 450008 Ufa, Russia
3 All-Russian Research Institute of Phytopathology, 143050 Bolshye Vyazemy, Russia
\* Correspondence: lira.gaisina@gmail.com

**Abstract:** Volcanic activity has a significant influence on the development of terrestrial ecosystems, including the Kamchatka Peninsula. We aimed to study the terrestrial algoflora of the Mutnovsky and Gorely volcanoes based on the use of clonal cultures of algae and cyanobacteria, and phenotypic and molecular genetic analyses. A total of 48 taxa were identified: 9 cyanobacteria, 32 Chlorophyta (11 Chlorophyceae, 21 Trebouxiophyceae), 3 Ochrophyta, and 4 Charophyta. In soils of the Mutnovsky volcano, 30 taxa were found, and in soils of the Gorely volcano, 24 were observed. In the studied area, small coccoid or mucilage-producing algae, which belong to cosmopolitan species, were identified, including representatives of the genera *Bracteacoccus*, *Chlorococcum*, *Coccomyxa*, *Coelastrella*, *Klebsormidium*, *Neocystis*, and *Vischeria*. Certain taxa were detected for the first time in the studied region, including *Bracteacoccus bullatus*, *Chlorococcum hypnosporum*, *Chlorococcum lobatum*, *Coccomyxa subellipsoidea*, *Klebsormidium nitens*, *Leptosira obovata*, *Lobosphaera incisa*, *Parietochloris pseudoalveolaris*, *Stenomitos tremulus*, and *Vischeria magna*. Our analysis of the algal communities at different altitudes reveals expansion in species richness with increasing distance from the tops of the volcanoes. The obtained data allowed us to estimate the real biodiversity of terrestrial algae and cyanobacteria of Kamchatkan volcanic soils, as well as the ecologies of these microorganisms.

**Keywords:** extreme habitat; molecular genetic analysis; strain; 16S rRNA; 18S rRNA; ITS; succession; algal communities; cosmopolitans

## 1. Introduction

A volcanic substrate is a lifeless space characterized by a small quantity of biogenic elements and direct exposure to abiotic factors, such as ultraviolet radiation and heating, precipitation, and wind. Microorganisms that live in such conditions must have a sufficient number of adaptive mechanisms for survival and further vital activity [1]. For example, it was long believed that cyanobacteria were the first to inhabit volcanic ash. There are many examples confirming this idea [2–6]. There is also an opinion that, together with cyanobacteria, liver mosses [7], green algae [8], or diatoms [9] may have been the first settlers of volcanic ash.

The texture (size of particles) and water-holding properties of ash, the presence of a constant flow of water, and a biotic component—a source of necessary biogenic elements—are factors that influence the process of colonization of ashes by microscopic cyanobacteria and algae. Observations of algal growth on coarser-grained pyroclastic material in areas of high rainfall near Mount St. Helens have shown that texture becomes a secondary factor with

constant water inflow [10]. As a result of laboratory modeling of the changing moisture regime, the hypothesis was confirmed that the appearance of predators (nematodes) contributes to an increase in the biomass of algae, since animals serve as an additional source of organic matter [11].

Volcanoes of the Earth belong to the mobile zones of the Earth's crust. The location of volcanoes within these zones is closely related to deep faults reaching the subcrustal region [12]. One of the first studies of the biota of the Krakatau volcano was undertaken in the Mediterranean–Indonesian zone (Malay Archipelago, Indonesia) [2–4], and the biota of Surtsey Iceland, related to the Atlantic zone (Iceland), was also examined [13–18]. The microbiota of volcanoes in the Pacific zone have been studied in detail on Deception Island (the South Shetland Islands archipelago, Antarctica) [19,20], in the Sokompa volcano (Chile) [21,22], at Mount St. Helens (Cascade Mountains, USA) [10], in the Katmai volcano (Alaska, USA) [23], in the Sierra Negra and Alcedo volcanoes (Galapagos Islands, Ecuador) [24], and in the Rincon de la Vieja volcano (Central America, Costa Rica) [25]. This zone also includes the volcanoes of the Kuril–Kamchatka volcanic belt: Golovnina, Tyatya, Mendeleeva, Tolbachik, Gorely, Mutnovsky, Avachinsky, and Shiveluch [26–34].

The Kamchatka Peninsula belongs to regions with very high volcanic activity [35]. The first study of cyanobacteria and algae of volcanic substrates in Russia was carried out on the Tyatya and Golovnin volcanoes (Kunashir Island, Kuril Islands) and on the Tolbachik volcano (Kamchatka Peninsula) after the Great Tolbachik Fissure Eruption (GTFE) [28]. The first four species of cyanobacteria were found around the release of volcanic vapor at temperatures of 50–70 °C in the Brave crater of the Tyatya volcano in 1980. Among them, *Mastigocladus laminosus* Cohn ex Kirchner was discovered, which is usually found in hot springs [27,36]. In total, 74 species of cyanobacteria and algae were found in the ashes, slags, and buried soils of the Tyatya, Golovnin, and GTFE volcanoes [26, 28]. In the soils of the Tolbachik volcano in Kamchatka, mainly green and yellow-green algae were found, occasionally some diatoms, and only one *Nostoc* sp. The species composition of algoflora was dominated by *Mychonastes homosphaera*, *Bracteacoccus minor*, *Pseudococcomyxa simplex*, *Myrmecia bisecta*, and *Stichococcus minor*. It was noted that in the studied volcanic soils of Kamchatka and the Kuril Islands, *Bracteacoccus minor* and *Pseudococcomyxa simplex* almost always participated in the development of the surface layer of ash. The small sizes of algae of all species were noted, the algae mainly falling on the substrate from the air and the so-called "relict algal flora" from the buried soil. A total of 20 algae and cyanobacteria species were recorded from the Goncharov and Pogibshaya lava tubes, located on the southeastern slope of the Gorely volcano [29].

It should be noted that in previous studies species identification was carried out using only a morphological approach which did not allow precise species identification.

The aim of the study was to investigate the species compositions of cyanobacteria and algae inhabiting volcanic soils of the seismically active Kamchatkan volcanoes Mutnovsky and Gorely using a polyphasic approach, which included clonal culture isolation, light microscopy, and molecular genetic analysis of the DNA of the studied microorganisms.

## 2. Materials and Methods

### 2.1. Study Site

The Mutnovsky volcano is one of the active volcanoes in the south of Kamchatka (Figure 1). The volcano is formed by four merged stratovolcanoes of Late Pleistocene–Holocene age. All of them are mainly composed of low-potassium and calc-alkaline basalts. The active Gorely volcano is related to the Mutnovsky geothermal area and is located 70 km from the city of Petropavlovsk-Kamchatsky and 25 km from the Pacific Ocean coast [37] (Figure 1). The volcano is a ridge about 7 km long, stretches in the west–northwest direction, and consists of five small stratovolcanoes merged with each other [38].

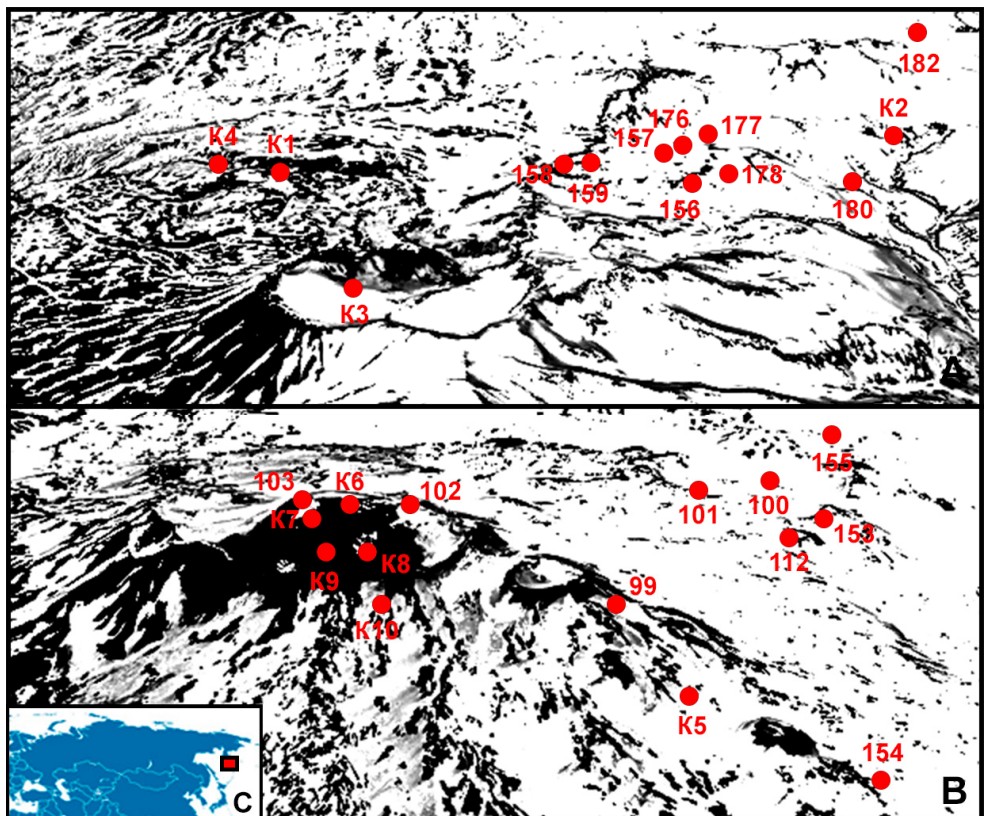

**Figure 1.** Study area. (**A**) Mutnovsky volcano. The red dots indicate the sampling sites. (**B**) Gorely volcano. (**C**) The red rectangle indicates the Kamchatka Peninsula. (According to [33], with modifications.)

The Mutnovsky (52°26′59.1″ N 158°11′42.7″ E) and Gorely (52°33′31.0″ N 158°02′16.0″ E) volcanoes are located in the southern volcanic district. The vegetation cover is characterized by a decrease in high-altitude vegetation belts. The upper boundary of the stone birch belt is located at an altitude of about 200 m above sea level. The wide development of the belt of alder and cedar dwarf trees at altitudes from 200 to 800 m is characteristic. Lava plateaus at altitudes of 800–1000 m are covered with mountain–tundra communities, among which communities of the *Rhododendron camtschaticum* Pall., *Phyllodoce aleutica* (Spreng.) A.Heller, *Phyllodoce caerullea* (L.) Bab. Nicholas lawns with the *Parageum caltifolium* (Menz.) Nakai et. Hara, and *Geranium erianthum* DC. are widely distributed [39].

The distribution of soils in Kamchatka has a well-defined altitudinal zonality, which is related to the patterns of altitudinal differentiation of the vegetation cover. The upper parts of the mountains, at altitudes above 1800 m, lack true soils. There are stone debris and placers and rocks of long-standing snowfields widespread here; some volcanoes and mountain ranges have glaciers [40].

A belt of mountain–tundra soils is located at altitudes less than 1700–1800 m. There are four subtypes of mountain–tundra soils developing in this belt: gley, non-gley humus, soddy permafrost, and volcanic layered ash. Mountain–tundra illuvial–humus soils dominate at altitudes of 1100–1500 m in the mountain–tundra belt, developing under lichen–shrub tundra, and mountain–tundra illuvial–humus volcanic destructive soils develop on a full ash column in the zone of moderate ash falls [41]. Mountain–tundra soils are characterized by base saturation up to 15–18% and pH values ranging from 4.8 to 5.0 [40].

Below 900–1100 m above sea level, there is a subalpine dwarf tree belt. In the elfin zone at altitudes of 700–1000 m, there are purulent–ocher soils (under alder forests), peaty illuvial–humus soils, and peaty illuvial–humus volcanic soils (under elfin pine forests). The latter are characterized by a complex polygenetic profile, consisting of several elementary

profiles with a thick organogenic horizon of a peaty nature, consisting of dry peaty remains of needles, the litter of shrubs, moss feathers, and lichens. In the lower horizons of such soils, especially in places with increased slope moisture, long seasonal permafrost is often observed [41,42]. The soils under dwarf pine trees are distinguished by a moderately acidic reaction of the environment: pH 4.7 in the surface humus horizons and 4.6 in humus [40].

For humus–humus soils under dwarf alder, a thin litter (1–3 cm) consisting of litter of alder and grasses is characteristic of high-humus horizon A1 (up to 80% loss during calcination in the surface humus horizons and up to 20–30% of humus in humus–humus) and gray-brown horizon B, where humus seeps from the upper horizon. The soils developing under the communities of dwarf alder are characterized by a moderately acidic reaction of the environment: pH 4.7 in the surface humus horizons and 4.6 in humus. The degree of saturation with bases in them is low [40].

The soils of the subalpine dwarf belt are distinguished by the peaty character of the modern organogenic horizon, the humus or semi-turfy nature of the buried organogenic horizons, the predominance of brown tones in the color of illuvial metamorphic horizons, the inhibition (in comparison with the forest zone) of the processes of weathering and accumulation of substances, high acidity, and unsaturation [41,43].

For the foothills and lower parts of the mountains, as well as high above the floodplain terraces of intermountain valleys within the Kamchatka mountain province at altitudes of 600–800 m, the formation of forest soils under woody vegetation is characteristic. In the forest belt (up to 600–800 m above sea level), in the zone of intense ash falls, layered ocher volcanic soils prevail, found under stone birch forests and, less often, white birch forests. They are characterized by the alternation of numerous buried elementary profiles.

## 2.2. Sample Collections

Samples of soil were collected in August 2010 (10 samples) and August 2020 (17 samples) (Table 1; Figures 1 and 2). Samples (no more than 500 g of soil) from each study site were taken with a metal spatula from the upper topsoil layer (<5 mm) according to the classic methods of soil phycology [44] and put into sterile paper bags.

**Table 1.** Sampling sites.

| Number | Description | Name | Year | GPS, Height above Sea Level | pH * | Soil Moisture, % * | Type of Soil * | Substrate Temperature |
|---|---|---|---|---|---|---|---|---|
| | | | | Mutnovsky volcano samples | | | | |
| 1 | Canyon of the Vulkannaya river, under the bushes | K1 | 2010 | 52°28′29.4″ N 158°06′47.8″ E, 858 m | 5.45 | 50–68 | Mountain–tundra illuvial–humus soils | 11 |
| 2 | At the base of the volcano, not far from Dachnye springs, alder forest | K2 | 2010 | 52°31′54.6″ N 158°11′55.0″ E, 773 m | 6.25 | 75–85 | Humus–ocher soils | 14 |
| 3 | 300 m from the top of the volcano | K3 | 2010 | 52°27′26.4″ N 158°09′50.4″ E, 1627 m | – | – | Stone talus and placers, rocks | 13 |
| 4 | In the lower part of the Vulkannaya River canyon | K4 | 2010 | 52°28′17.3″ N 158°06′02.4″ E, 739 m | – | – | Rocks | 13 |
| 5 | Clump of sedge on the crumbling southern slope | 156 | 2020 | 52°31′20.7″ N 158°09′91.1″ E, 1053 m | 4.34 | 55 | Mountain–tundra sod frozen | 11 |
| 6 | Alpine meadow | 157 | 2020 | 52°30′74.4″ N 158°09′90.2″ E, 1039 m | 4.74 | 75 | Mountain–tundra illuvial–humus soils | 10 |
| 7 | Willow curtain | 158 | 2020 | 52°30′08.7″ N 158°09′50.7″ E, 1145 m | 4.23 | 65 | Mountain–tundra sod frozen | 11 |
| 8 | Alpine meadow dominated by willow and legumes | 159 | 2020 | 52°29′95.2″ N 158°09′29.4″ E, 1193 m | 4.52 | 68 | Mountain–tundra illuvial–humus soils | 10 |

<div align="center">**Table 1.** *Cont.*</div>

| Number | Description | Name | Year | GPS, Height above Sea Level | pH * | Soil Moisture, % * | Type of Soil * | Substrate Temperature |
|---|---|---|---|---|---|---|---|---|
| 9 | Alpine meadow | 176 | 2020 | 52°31′12.8″ N 158°09′81.2″ E, 1065 m | 5.62 | 70 | Mountain–tundra illuvial–humus soils | 11 |
| 10 | Alder Dwarf Curtain | 177 | 2020 | 52°31′03.9″ N 158°09′74.5″ E, 1067 m | 5.15 | 78 | Mountain–tundra illuvial–humus soils | 13 |
| 11 | Forbs with the dominance of wormwood, willow on the slope of the stream | 178 | 2020 | 52°30′93.4″ N 158°10′34.0″ E, 945 m | 5.05 | 65 | Tundra volcanic illuvial–humus soils | 11 |
| 12 | Alpine forb meadow on a volcanic plateau near the Mutnovskaya geothermal station | 182 | 2020 | 52°33′87.0″ N 158°11′22.0″ E, 885 m | 5.36 | 73 | Mountain–tundra illuvial–humus soils | 13 |
| | | | | Gorely volcano samples | | | | |
| 1 | Slope, flat area among sedges | K5 | 2010 | 52°32′46.2″ N 158°02′39.9″ E, 1501 m | 4.35 | 55 | Illuvial–humus volcanic destructive soils | 12 |
| 2 | The trail along the edge of the crater, green layer on the surface of the ground | K6 | 2010 | 52°33′26.4″ N 158°02′09.2″ E, 1758 m | 5.68 | 45 | Volcanic ash, sand | 18 |
| 3 | Down the east slope | K7 | 2010 | 52°33′19.1″ N 158°01′57.4″ E, 1690 m | 5.15 | 55 | Tundra volcanic illuvial–humus soils | 14 |
| 4 | At the edge of a crater with a lake | K8 | 2010 | 52°33′12.8″ N 158°02′20.7″ E, 1675 m | 5.25 | 45 | Sulfur deposits around the crater | 16 |
| 5 | Down the east slope | K9 | 2010 | 52°33′10.8″ N 158°02′06.0″ E, 1645 m | 4.65 | 55 | Tundra volcanic illuvial–humus soils | 12 |
| 6 | Down the east slope | K10 | 2010 | 52°32′53.7″ N 158°02′21.6″ E, 1555 m | 4.55 | 55 | Tundra volcanic illuvial–humus soils | 12 |
| 10 | Eastern slope under the snowfield | 99 | 2020 | 52°34′26.7″ N 158°04′92.9″E, 1060 m | 4.35 | 65 | Mountain–tundra sod frozen | 10 |
| 11 | Solidified lava flow on the eastern slope | 100 | 2020 | 52°34′07.9″ N 158°04′92.9″ E, 1192 m | 4.48 | 60 | Mountain–tundra sod frozen | 11 |
| 12 | Dead bean clump | 101 | 2020 | 52°33′91.3″ N 158°04′35.9″ E, 1310 m | 4.72 | 57 | Mountain–tundra sod frozen | 12 |
| 13 | Scree of soil under the rock | 102 | 2020 | 52°33′53.1″ N 158°02′32.5″ E, 1784 m | – | – | Rocks | 14 |
| 14 | Thermal vapor outlet along the edge of the caldera | 103 | 2020 | 52°33′30.6″ N 158°01′74.2″ E, 1805 m | – | – | Rocks | 32 |
| 15 | Dead clump of sedge on a lava terrace | 112 | 2020 | 52°33′67.4″ N 158°05′14.1″ E, 1226 m | 4.47 | 45 | Mountain–tundra sod frozen | 10 |
| 16 | The vent of a small side crater filled with lava chips | 153 | 2020 | 52°33′75.4″ N 158°05′33.7″ E, 1207 m | 4.82 | 56 | Mountain–tundra sod frozen | 11 |
| 17 | Alpine meadow at the foot of the volcano | 154 | 2020 | 52°34′27.6″ N 158°05′46.1″ E, 1064 m | 4.76 | 75 | Mountain–tundra illuvial–humus soils | 10 |
| 18 | Old overgrown alluvial cone overgrown with sedges | 155 | 2020 | 52°34′65.0″ N 158°05′30.6″ E, 1002 m | 4.35 | 65 | Mountain–tundra sod frozen | 10 |

Notes: * According to [39–43].

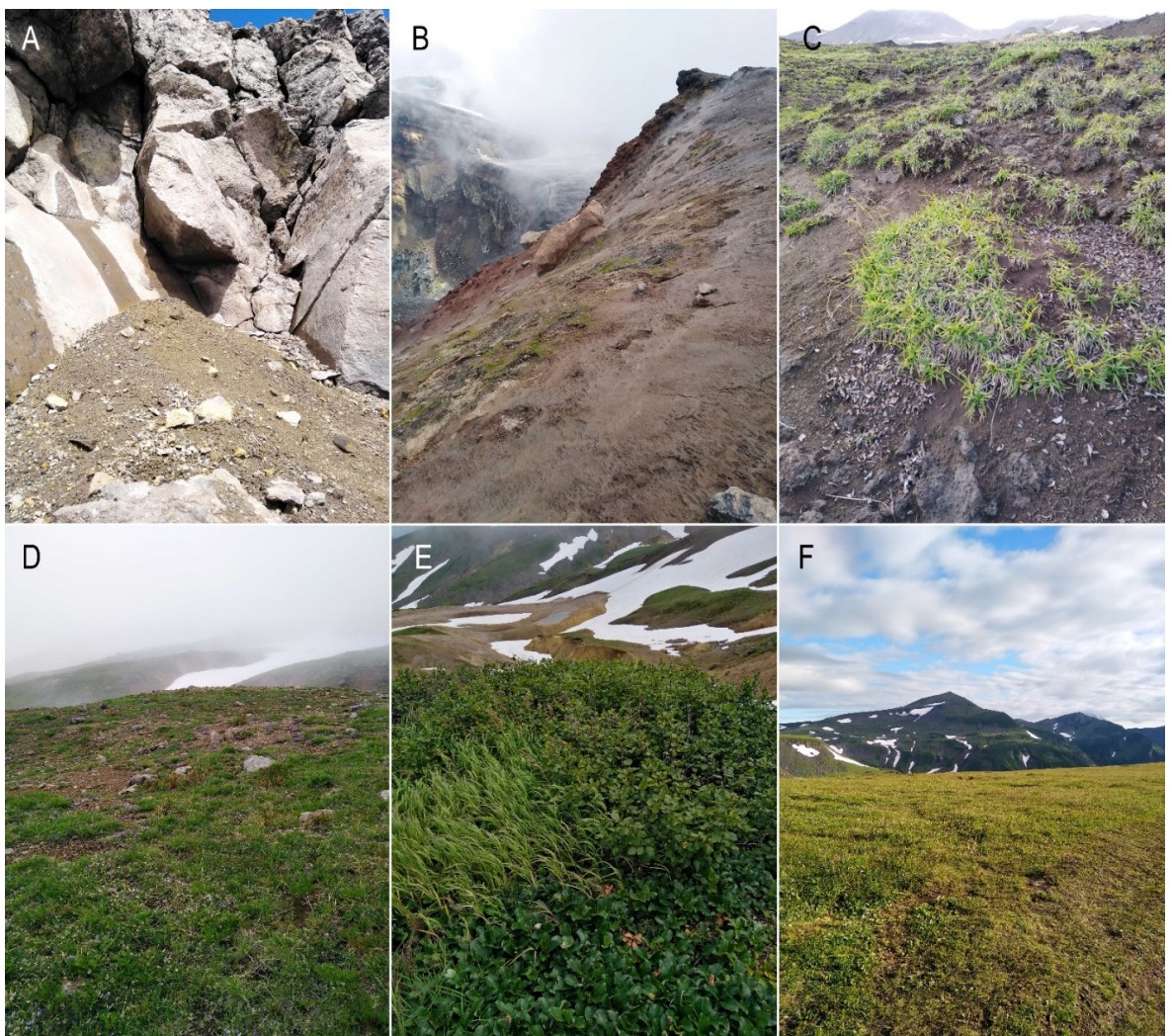

**Figure 2.** Study sites: (**A**) 102, scree of soil under the rock of Gorely volcano; (**B**) 103, thermal vapor outlet along the edge of the caldera on Gorely volcano; (**C**) 112, dead clump of sedge on a lava terrace on Gorely volcano; (**D**) 176, alpine meadow on Mutnovsky volcano; (**E**) 177, alder dwarf curtain on Mutnovsky volcano; (**F**) 182, alpine forb meadow on a volcanic plateau near the Mutnovskaya geothermal station on Mutnovsky volcano.

During sampling, the sample number, height above sea level, and coordinates (geoposition) were recorded and a description of the area (habitat, nature of the substrate, and vegetation) was given. The sampling site and the surface layer of the soil were photographed. The soil temperature was measured with a Steklopribor TB-3-M1 thermometer.

Air-dried samples were stored at room temperature in the laboratory in a dark place. Collected material was used for the establishment of enrichment cultures within three weeks after the sampling.

### 2.3. Cultivation of Strains and Morphological Identification

For the identification of photosynthetic organisms inhabiting volcanic soils, an integrative approach was applied. Small amounts of soil sample (approximately 1 g) were inoculated in liquid 3N BBM [45], Z8 [46], and Waris-H [47] media to promote the growth of cyanobacteria and eukaryotic algae.

Enrichment cultures were incubated at room temperature with a photon fluence of 17.9–21.4 µmol photons·m$^{-2}$ s$^{-1}$ in a 16:8 h light/dark cycle. After the 4-week cultivation, single cells of algae or filaments of cyanobacteria could be observed, which were then

transferred to new plates to establish pure clonal cultures; the unialgal or unicyanobacterial cultures were incubated under the same conditions.

The strains were isolated using the micropipette method and dilution technique [45] and cultured in liquid nutrient media and agar slant in tubes.

The strains were maintained in the Bashkortostan Collection of Algae and Cyanobacteria (BCAC) (WDCM 1023, Ufa, Russia) and the culture collection of the Laboratory of Botany in the Federal Scientific Center of East Asia Terrestrial Biodiversity (Vladivostok, Russia).

The morphology of the strains was examined with an Axio Imager A2 (Carl Zeiss, Oberkochen, Germany) and an Olympus BX 53 (Olympus, Tokyo, Japan), equipped with Nomarski DIC optics. Microphotographs were taken with Axio Cam MRC (Carl Zeiss, Germany) and Olympus DP27 (Olympus, Japan) cameras at ×1000 magnification.

For the identification of algae and cyanobacteria, relevant references [48–54] and recent publications [55,56] were used.

### 2.4. DNA Extraction, PCR

Selected strains of algae and cyanobacteria were studied using molecular genetic methods. Cultures were harvested during the exponential growth phase and concentrated by centrifugation. Total genomic DNA was extracted as described previously [57]. For the Chlorophyta members, SSU rDNA and the ITS region of rDNA were amplified using primer combinations and temperature profiles, following Nemcová et al. [58] and Gontcharov et al. [59]. The amplification and sequencing of the ITS region of members of Eustigmatophyceae were conducted as described previously [60], with modifications. The amplification of the cyanobacterial 16S rRNA gene and the 16S–23S ITS region was performed using primers and temperature profiles described previously [57]. PCR amplification was performed using the Encyclo Plus PCR kit (Evrogen, Moscow, Russia) with a T100 Thermal Cycler (Bio-Rad Laboratories, Inc., USA). The PCR products were purified by ExoSAP-IT PCR Product Cleanup Reagent (Affymetrix Inc., USA) and sequenced in both directions at the FSCEATB FEB RAS using an ABI 3500 genetic analyzer (Applied Biosystems, USA) with a BigDye terminator v. 3.1 sequencing kit (Applied Biosystems, Maryland, USA). Sequences were assembled with the Staden Package v.1.4 [61] and aligned manually in the SeaView program [62].

The sequences of the 18S rRNA gene and ITS region of eukaryotic algae and the 16S rRNA gene and the 16S–23S ITS region of cyanobacteria were compared with those from authentic and references strains available at the National Center for Biotechnology Information (NCBI, Bethesda, USA) by means of a BLAST search (https://blast.ncbi.nlm.nih.gov/Blast.cgi; accessed on 20 September 2021) for estimation of their taxonomic position. In the case of 99–100% similarity with sequences from NCBI, the identity of cyanobacteria and algae at the species level was assumed; with a similarity of 95–98%, the identity of organisms on the genus level was assumed.

### 2.5. Phylogenetic Analysis

A dataset used to access the affinity of our *Coelastrella* isolates was assembled using ITS rRNA sequences of the genus representatives (≥95% similarity according to the BLAST search) and related taxa retrieved from the NCBI. The sequences (taxa names and accession and strain numbers are given as listed in the NCBI) were aligned in the SeaView program [62] with manual corrections. The final alignment consisted of 84 sequences with 626 aligned base positions.

The best evolutionary model GTR+I+G was determined using jModelTest 2.1.1 [63]. Phylogenetic trees were constructed using the maximum likelihood (ML) method in RAxML v.7.2.6 (http://embnet.vital-it.ch/raxml-bb/; accessed on 1 April 2022) [64] and Bayesian inference (BI) in MrBayes v.3.1.2 [65]. In BI, four runs of four Markov chains were executed for 2 million generations, sampling every 100 generations for a total of 20,000 samples. Convergence of the chains was assessed, and stationarity was determined according to the "sump" plot, with the first 5000 samples (25%) discarded as burn-in; posterior probabilities

were calculated from trees sampled during the stationary phase. The robustness of the ML trees was estimated by examining the bootstrap percentages (BPs) [66] and posterior probabilities (PPs) in BI. Those with BPs <50% and PPs <0.95 were not considered.

## 3. Results

During the study of the Mutnovsky and Gorely volcanic soils, 48 taxa were detected: 9 cyanobacteria, 32 Chlorophyta (11 Chlorophyceae, 21 Trebouxiophyceae), 3 Ochrophyta, and 4 Charophyta (Table 2, Figure 3A–AV). Identification of most of the taxa was based on a combination of morphological and molecular genetic approaches. In soils of the Mutnovsky volcano, 30 taxa were found, and in soils of the Gorely volcano, 24 were found.

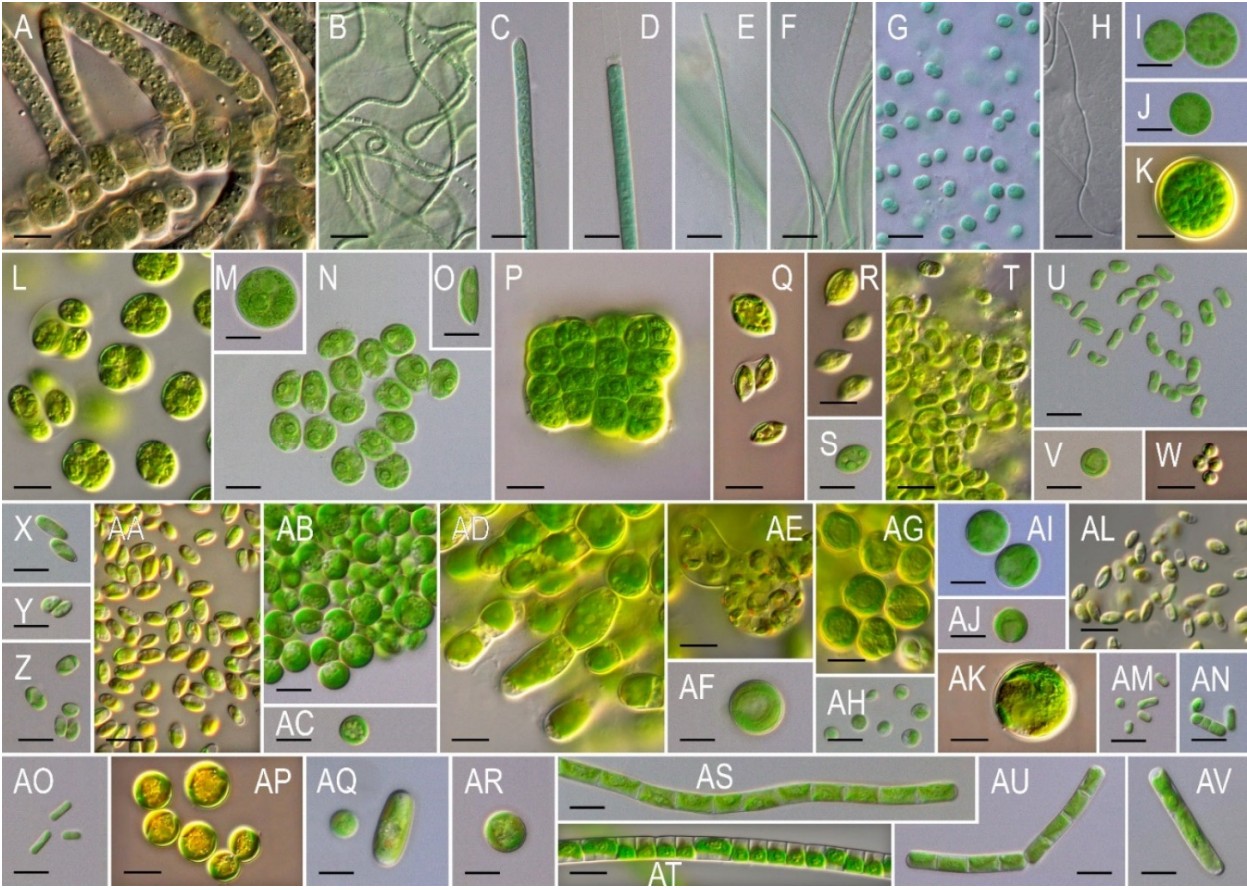

**Figure 3.** Algae and cyanobacteria from Mutnovsky and Gorely volcano soils: (**A**) *Fischerella* cf. *major*; (**B**) *Leptolyngbya* cf. *foveolarum*; (**C**) *Microcoleus* cf. *calidus*; (**D**) *Phormidium* cf. *corium*; (**E**) *Stenomitos tremulus*; (**F**) *Stenomitos* sp; (**G**) *Synechocystis* cf. *salina*; (**H**) cf. *Trichocoleus hospitus*; (**I**) *Bracteacoccus bullatus*; (**J**) *Bracteacoccus minor*; (**K**) *Bracteacoccus* sp. 1; (**L**) *Chlamydocapsa* cf. *lobata*; (**M**) *Chlorococcum hypnosporum*; (**N**) *Chlorococcum lobatum*; (**O**) *Chlorolobion* cf. *lunulatum*; (**P**) *Chlorosarcinopsis* sp.; (**Q**) *Coelastrella aeroterrestrica*, strain C_aero; (**R**) *Coelastrella oocystiformis*, surface view; (**S**) *Coelastrella terrestris*, median view; (**T**) *Neocystis mucosa*, strain K2 N_muc; (**U**) *Neocystis mucosa*, strain 1272; (**V**) *Chlorella* cf. *chlorelloides*; (**W**) *Chlorella* sp. 2; (**X**) *Coccomyxa subellipsoidea*; (**Y**) cf. *Coccomyxa viridis*; (**Z**) *Coccomyxa* sp. 2, strain 1237; (**AA**) *Coccomyxa* sp. 1; (**AB**) *Elliptochloris* cf. *reniformis*; (**AC**) *Elliptochloris* cf. *subsphaerica*; (**AD**) *Leptosira obovata*, vegetative cells; (**AE**) *Leptosira obovata*, zoosporangium; (**AF**) *Lobosphaera incisa*, strain 1248; (**AG**) *Lobosphaera* sp.; (**AH**) *Micractinium* sp; (**AI**) *Myrmecia* sp. 1; (**AJ**) *Parietochloris pseudoalveolaris*, strain 1306; (**AK**) *Parietochloris* sp.; (**AL**) *Pseudococcomyxa* sp.; (**AM**) *Stichococcus* sp. 1; (**AN**) *Stichococcus* sp. 2; (**AO**) *Stichococcus* sp. 3; (**AP**) *Vischeria magna*; (**AQ**) *Vischeria* cf. *stellata*; (**AR**) *Vischeria* sp.; (**AS**) *Klebsormidium nitens*; (**AT**) *Klebsormidium* sp. 1; (**AU**) *Klebsormidium* sp. 2; (**AV**) *Mesotaenium* sp. Scale bar: 10 μm.

In the samples K4, 159, and 156 from the Mutnovsky volcano, 10, 8, and 7 taxa were detected, respectively (Table 2). In the Gorely volcano, the most diverse in algae and cyanobacteria was sample 112, with six taxa. In other samples from both volcanoes, only one to five species were identified. The average number of taxa per sample was 3.33, which corresponds to relatively low biodiversity.

Some taxa were represented by several strains: *Coelastrella aeroterrestrica* had five strains; *Neocystis mucosa,* three strains; *Coccomyxa subellipsoidea,* three strains; and *Parietochloris pseudoalveolaris,* two strains.

The representatives of Chlorophyta were the most diverse in the studied area. With respect to species composition, coccoid cosmopolitan and widespread genera prevailed, including *Bracteacoccus*, *Coccomyxa*, *Coelastrella*, *Neocystis*, and *Chlorococcum*.

In soils of both volcanoes, *Chlorococcum hypnosporum*, *Chlorolobion* cf. *lunulatum*, *Coccomyxa subellipsoidea*, *Coelastrella aeroterrestrica*, *Coelastrella terrestris*, *Elliptochloris* cf. *reniformis*, *Elliptochloris* cf. *subsphaerica*, *Eremochloris kamchatica*, *Leptolyngbya* cf. *foveolarum*, *Neocystis mucosa*, and *Vischeria magna* were detected.

Among cyanobacteria, filamentous species dominated. Some strains were identified based only on morphological characters. An interesting finding was *Fischerella* cf. *major*. It has erect cylindrical branches 6–12 µm wide. Sheaths are thin, sometimes invisible. Trichomes consist of almost quadratic cells, constricted at the cross-walls, 7–9 µm wide. Cells are cylindrical, 6–8 µm wide, in barrel-shaped, isodiametric branches. Heterocytes are intercalary and cylindrical. Hormogonia and akinetes are not observed (Figure 3A).

*Microcoleus* cf. *calidus* was identified in sample 103 near the thermal steam outlet along the edge of the southern slope of the Gorely volcano caldera (Figure 2B) at a ground temperature of 32 °C (Table 1) and was morphologically similar to other members of the genus *Microcoleus* (Figure 3C). Molecular genetic analysis of 16S-23S ITS and 16S rRNA revealed that the strain was very similar to the strain DAI *Microcoleus* sp. (Table 2). Based on morphological, molecular genetic, and ecological data, the strain was identified as *Microcoleus* cf. *calidus*.

A strain of cyanobacteria (K7) was isolated from a sample (K9) from the eastern slope of the tundra volcanic illuvial–humus soil (Table 1), characterized by trichomes tapering to the ends with false branching and the presence of heterocysts. Analysis of the 16S rRNA gene revealed that the strain belongs to a new species of *Roholtiella*. These results will be published in a separate paper.

Strain 1296, *Stenomitos tremulus*, was detected based on morphology and molecular genetic data (Table 2, Figure 3E). It has thin trichomes (1.1–1.8 µm wide), a cell length of 1.5–2.8 µm, and conical or slightly rounded end cells. The 16S rRNA sequence of this strain was almost identical to the sequence of strain UTCC 471, *Stenomitos tremulus* (Table 2). This taxon was described in freshwater habitats [67]. The discovery of *Stenomitos tremulus* in volcanic soils demonstrated the ecological plasticity of the species and its ability to survive in extreme terrestrial conditions.

The first record of the cyanobacteria *Synechocystis* cf. *salina*, which is common in brackish water ecosystems [52], expands our knowledge of the ecology of this species. Cells of *Synechocystis* cf. *salina* are bright blue-green, with granular cytoplasm, solitary or in pairs, spherical and hemispherical, and 2–3 µm in diameter (Figure 3G).

In Mutnovsky and Gorely volcanic soils, Chlorophyta constituted the most diverse group. Several strains of *Bracteacoccus* were isolated (Table 2, Figure 3I–K). Two strains were investigated by molecular genetic methods. The sequence of the ITS region of the strain 1366 was almost identical to the sequence of the authentic strain SAG 2032 *Bracteacoccus bullatus* (Table 2). The sequence of the ITS region of the strain 1228 was characterized by 99.41% similarity with the sequence of strain TTF-2-1-J *Bracteacoccus minor* (Table 2). It should be noted that, morphologically, the strains were very similar, with spherical cells of 5–32 diameter with numerous chloroplasts (Figure 3I–K). For some *Bracteacoccus* strains (K2, K3, K4, K5, K10), the molecular data were not obtained due to unsuccessful PCR

amplification of marker regions, so they were identified only at the genus level. The same situation is noted for some strains below.

*Chlamydocapsa* cf. *lobata* was recognized based on morphology only (Table 2, Figure 3L). The cells were in colonial lamellate mucilage, spherical or almost spherical, and 9–14 μm in diameter. The chloroplast was bladed, with a single pyrenoid.

During the investigations, two strains (1264 and 1269) (Table 2; Figure 3M,N), morphologically similar to *Chlorococcum*, were isolated. The cells are ovoid or spherical, 9–30 μm in length, 7–10 μm in width, and a single chloroplast contains one pyrenoid. The ITS sequence of the strain 1269 was almost (percent) identical to the sequence of the authentic strain SAG 213-6 *Chlorococcum hypnosporum*, while strain 1264 was very similar to the sequence of SAG 12.84 *Chlorococcum lobatum*. Therefore, we identified them as *Chlorococcum hypnosporum* and *Chlorococcum lobatum*, respectively.

*Chlorolobion* cf. *lunulatum* and *Chlorosarcinopsis* sp. were identified using only morphology (Table 2; Figure 3O,P). Cells of *Chlorolobion* cf. *lunulatum* were elongate, lunate, attenuated at the apices, with an elongated chloroplast, without a pyrenoid (Figure 3O). Cells of *Chlorosarcinopsis* sp. were in packets, 6–11 μm in diameter, with a massive chloroplast with a single pyrenoid (Figure 3P).

Representatives of the Scenedesmaceae family were very frequent in soils of the Mutnovsky and Gorely volcanoes (Table 2). Six strains of algae, similar to *Coelastrella* genus, were isolated (Figure 3Q–S). The morphology of these strains was also similar. They have lemon-like cells, 8–16 μm in length, 4–10 μm in width, without polar thickenings, and cup-shaped chloroplasts with a parietal or central pyrenoid. Phylogenetic analysis revealed that five strains (K10, 1231, 1234, 1236, 1260) were almost identical to the authentic strain SWK1_2 *Coelastrella aeroterrestrica* (Figure 4, Table 2). Strain K1 *Coelastrella oocystiformis* was similar to authentic strain SAG 277-1 *Coelastrella oocystiformis* (Figures 4 and 3R,S; Table 2). It is known that representatives of the Scenedesmaceae family are widely distributed in terrestrial ecosystems [27,68–72].

In the studied area, taxa from the genus *Neocystis* were distributed. Members of this group are characterized by kidney-like cells without pyrenoids surrounded by mucilage. In addition to strains K1 and K2 (Figure 3T), which were isolated from Mutnovsky volcanic soils in 2010 [34], the strain 1272 was isolated from the soils of this volcano in 2020 (Table 2, Figure 3U). The ITS sequences of all isolated *Neocystis* strains were close to the sequence of the strain SAG 40.88 *Neocystis mucosa*.

*Chlorella* cf. *chlorelloides* (Table 2, Figure 3V), isolated from a sample of an alpine meadow at the foot of the Gorely volcano (Table 1), differed from typical representatives of the genus with the absence of a pyrenoid and a parietal chloroplast lining almost the entire inner part of the cell.

Several strains of algae with *Coccomyxa*/*Pseudococcomixa*-like morphologies were isolated, with elliptic cells without pyrenoids in mucilage (Table 2; Figure 3X–AA,AL). Molecular genetic analysis revealed that three strains belong to the *Coccomyxa subellipsoidea*, as the sequences of the ITS region of strains 1249 and 1245 were very similar to the ITS sequence of the strain SAG 216-13 *Coccomyxa subellipsoidea*, and the ITS sequence of the strain 1271 was close to the sequence of the strain P6065 *Coccomyxa subellipsoidea* (Table 2, Figure 3X). Strain *Coccomyxa* sp. 2 1237 was identified using morphological characters, and the similarity of its ITS sequence to the sequences of the strains Obi *Coccomyxa* sp 2. (Table 2, Figure 3Z), *Coccomyxa* sp. 1 (Figure 3AA), and cf. *Coccomyxa viridis* (Figure 3Y) was detected only on the basis of morphology.

Two species of *Elliptochloris* were found: *Elliptochloris* cf. *subsphaerica* (Table 2, Figure 3AB) and *Elliptochloris* cf. *reniformis* (Table 2, Figure 3AC). Algae from this genus have round cells with single chloroplasts divided into two parts, and the cytoplasm is very granulated. *Elliptochloris* cf. *subsphaerica* differs from *Elliptochloris* cf. *reniformis* due to the presence of a pyrenoid [50]. The ITS sequence of *Elliptochloris* cf. *subsphaerica* (strain 1245) was similar to the sequence of the authentic strain CAUP H7101 *Elliptochloris subsphaerica*. The ITS

sequence of *Elliptochloris* cf. *reniformis* was closed to the sequence of the authentic strain of CAUP H7102 *Elliptochloris reniformis*.

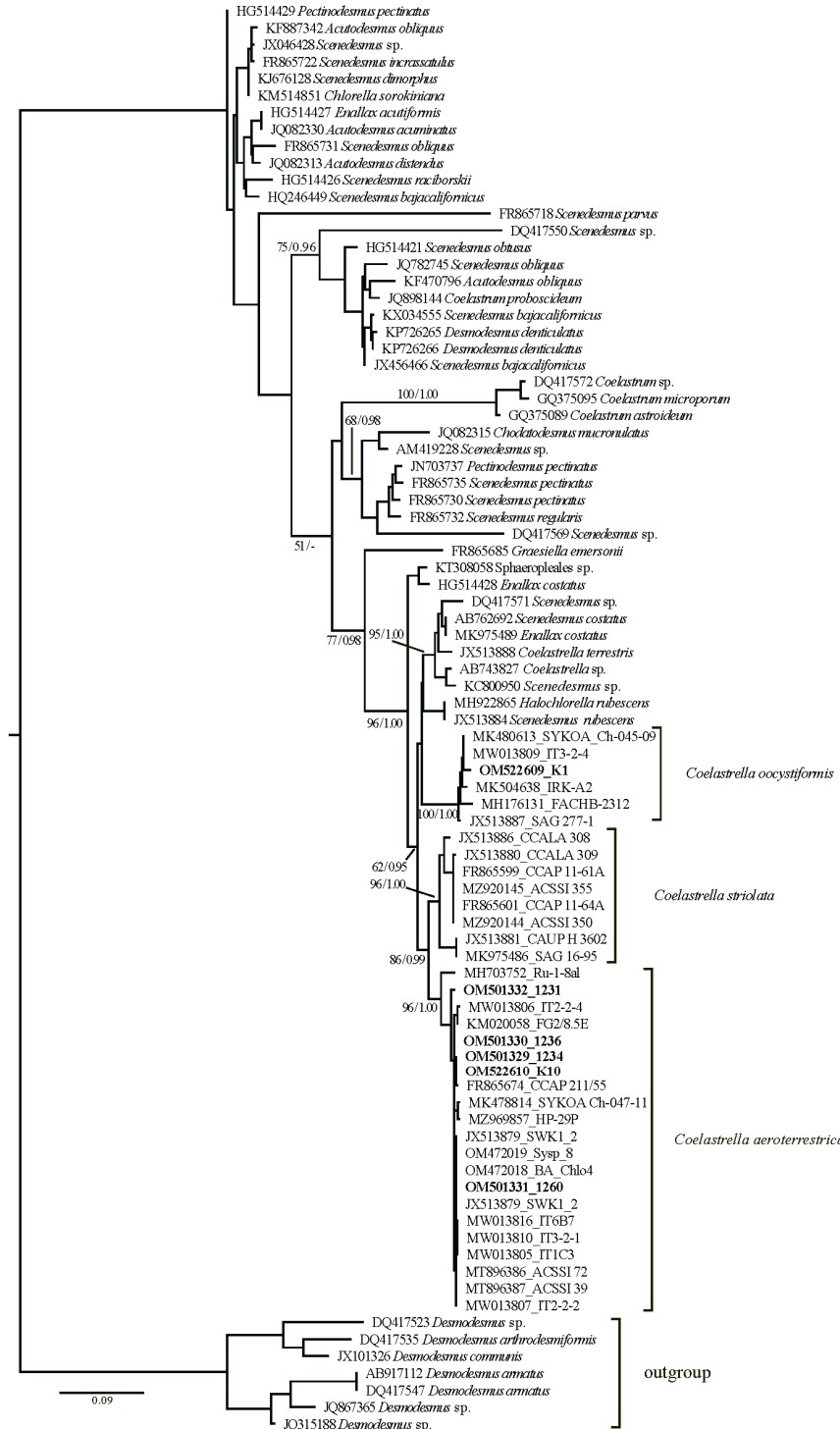

**Figure 4.** ML phylogenetic tree inferred in RAxML with a GTR+I+G nucleotide substitution model using 84 ITS rRNA sequences (626 characters). ML BP (>50%) and BI PP (>0.95) are shown. The newly obtained sequences are given in bold. Scale bar: substitutions per nucleotide position.

During the study, several strains of green algae, similar to representatives of genus *Eremochloris*, were isolated (Table 2). Molecular genetic analysis of the 18S rRNA genes and the ITS region confirmed that the strains belong to the new species of the genus *Eremochloris*–*Eremochloris kamchatica* [73].

Among the most interesting isolates was a strain morphologically similar to algae of the genus *Leptosira* (Figure 3AD,AE). On agar, the algae form bushes with branched filaments, with elongated, ellipsoid, or almost spherical cells that are 11.6–23.5 μm long and 9.4–15.9 μm wide. Terminal cells are mostly elongated. The chloroplast is parietal, with a naked pyrenoid. Zoosporangia are large, up to 23.3 μm in diameter. Zoospores are released in the mucous bladder and are spherical after release, 5–5.7 μm in diameter, with two flagella of equal length and a large stigma 2–2.15 μm in diameter. For establishing the phylogenetic position of the strain, the 18S rRNA gene was sequenced. The resulting sequence was almost identical to that of the authentic strain SAG 445-1 *Leptosira obovata* (Table 2). Thus, the newly isolated strain could be attributed to the species *Leptosira obovata*.

Another interesting taxon was the strain K9 *Lobosphaera* sp. (Table 2, Figure 3AG). The sequence of the ITS gene was closest to that of the authentic strain CAUP H 4301 *Lobosphaera incisa*. Morphologically, this species had features typical of the members of the genus *Lobosphaera*: cells were solitary or in clusters, 6.5–13.6 μm in diameter, a parietal chloroplast without a pyrenoid, divided into two hemispheres, a large nucleus, and cellular cytoplasm [51]. The formation of autospores was observed, and zoospores were not found. Despite the morphological similarity to *Lobosphaera incisa*, differences in the ITS sequence of the K9 strain *Lobosphaera* sp. were significant. Perhaps this strain is a new species of the *Lobosphaera* genus.

Strain *Micractinium* sp. was found at the outlet of thermal vapors along the rim of the Gorely volcano caldera (Figure 3H). Its 18S rDNA gene sequence has 99.9% similarity to the strain ACSSI 332 *Micractinium* sp., which was isolated from a freshwater reservoir (a hot spring) in Chukotka (Table 2). Algae have hemispherical or oval cells with a diameter of 2.5–5.0 μm and a single cup-shaped chloroplast with a pyrenoid.

*Myrmecia* sp. strain 1363 is typical for this genus morphology: spherical cells 7–11 μm in diameter, with one chloroplast without a pyrenoid (Figure 3I). The ITS sequence of this strain was almost identical to the sequence of Ru-s-3-3 *Myrmecia* sp.

In soils of both studied volcanoes, representatives of genus *Parietochloris* were identified (Table 2, Figure 3AI–AK). These algae have spherical cells 9–15 μm in diameter. The cup-shaped pyrenoid contains numerous starch grains. The ITS sequences of the strains 1289 and 1306, isolated in 2020, were almost identical to the sequence of NV-5 *Ettlia pseudoalveolaris* (the synonym of *Parietochloris pseudoalveolaris*). These strains were identified as *Parietochloris pseudoalveolaris* (Table 2, Figure 3AJ). *Parietochloris sp.* is also typical of the genus morphology, but without molecular genetic analysis it was identified only at the genus level (Table 2, Figure 3AK).

In the soils of the Gorely volcano, three species of *Stichococcus* were found (Table 2, Figure 3AM–AO). The morphology of all the strains was typical for the *Stichococcus* genus: short cylindrical cells with a slightly visible pyrenoid. ITS sequences of *Stichococus* sp. 2 and *Stichococus* sp. 3 (strains 1270 and 1286, respectively) (Figure 3AN,AO) have rather low similarities—88.44–89.86%—to the sequence of the *Stichococcus antarcticus* A. Beck FiSo15/03cVI (M) M-0019691.

Three algae from the phylum Ochrophyta were found very often in the samples from 2010 and 2020. These are representatives of the genus *Vischeria* (Table 2): *Visheria magna* (Figure 3AP), *Visheria* cf. *stellata* (Figure 3AQ), and *Vischeria* sp. (Figure 3AR). The sequence of the ITS region of the K9–10 *Vischeria magna* strain had a similarity of 97.44% with the authentic strain SAG 2554 *Vischeria magna*.

In the studied area, Charophyta accounted for four taxa. Several strains of *Klebsormidium* were identified (Table 2, Figure 3AS–AU). Molecular genetic analysis revealed that strain 1290 was almost identical to SAG 335-1a *Klebsormidium nitens* (Table 2, Figure 3AS). The morphology of the strain 1290 was typical of the genus *Klebsormidium*, with cells 8–20 μm long and 5–7 μm wide. The chloroplast is girdle-shaped or parietal, with one pyrenoid, located in the middle of the chloroplast. Two other strains, *Klebsormidium* sp. 1 and *Klebsormidium* sp. 2, were very similar to strain 1290, but on the basis of morphology alone, we could clearly identify only the genus.

*Mesotaenium* sp. was detected in the sample of an alpine forb meadow (182) (Figure 3F) on the Mutnovsky volcano. This genus usually prefers humid habitats. Its finding can be explained by the increased soil moisture under the alpine meadows, which was caused by high precipitation due to the melting of snowfields above the sampling site.

**Table 2.** List of algae and cyanobacteria in Mutnovsky and Gorely volcano soils.

| Taxa, Strain | Mutnovsky Volcano | | Gorely Volcano | | Genes, Percentage of Identity with Reference Strain, Accession Number of the Reference Strain | GenBank Accession Number, Publication Information |
|---|---|---|---|---|---|---|
| | 2010 | 2020 | 2010 | 2020 | | |
| Cyanobacteria | | | | | | |
| *Fischerella* cf. *major* Gomont | K4 ** | | | | | |
| *Leptolyngbya* cf. *foveolarum* (Rabenhorst ex Gomont) Anagnostidis et Komárek | K4 | | K8, K10 | | | |
| *Microcoleus* cf. *calidus* (Gomont ex Gomont) Strunecky, Komárek & J.R.Johansen *, strain 1267 | | | | 103 | 16S, 16S-23S rRNA, 96.44% identity with *Microcoleus* sp. DAI, EF654029 | OM501356 |
| *Phormidium* cf. *corium* Gomont ex Gomont | | | | 153 | | |
| *Roholtiella* sp. *, strain K7 | | | K9 | | 16-23S ITS rRNA, 97.71%, *Roholtiella bashkiriorum* RU9, KM268886 | |
| *Stenomitos tremulus* (J.R.Johansen & Casamatta) Miscoe & J.R.Johansen *, strain 1268 | | 156 | | | 16S rRNA, 99.23% identity with *Stenomitos tremulus* UTCC 471, AF218371 | OM501358 |
| *Stenomitos* sp. *, strain 1317 | | | | 99 | 16S-23S ITS rRNA, 96.69% identity with *Stenomitos* sp. WJT24NPBG20_P25, KF761557 | OM501357 |
| *Synechocystis* cf. *salina* Wislouch | | 159 | | | | |
| cf. *Trichocoleus hospitus* (Hansgirg ex Gomont) Anagnostidis | K1 | | | | | |
| Chlorophyta Chlorophyceae | | | | | | |
| *Bracteacoccus bullatus* Fučíková, Flechtner & Lewis *, strain 1366 | | 182 | | | ITS rRNA, 98.00% identity with *Bracteacoccus bullatus* SAG 2032, JQ281848 | OM501335 |
| *Bracteacoccus minor* (Schmidle ex Chodat) Petrová *, strain 1228 | | | | 99–101 * | ITS, 18 S rRNA, 99.41% identity with *Bracteacoccus minor* TTF-2-1-J, MT991535 | OM501328 |
| *Bracteacoccus* sp.1 | K2, K4, | | K5, K10 | | | |
| *Chlamydocapsa* cf. *lobata* Broady | K4 | | | | | |
| *Chlorococcum hypnosporum* Starr *, strain 1269 | | | | 102, 112 | 18S, ITS rRNA, 99.80% identity with authentic strain *Chlorococcum hypnosporum* SAG 213-6, JN904003 | OM501336 |
| *Chlorococcum lobatum* (Korshikov) F.E.Fritsch & R.P.John *, strain 1264 | | 157 | | | 18S, ITS rRNA 99.20% identity with *Chlorococcum lobatum* SAG 12.84, AB936289 | OM501352 |
| *Chlorolobion* cf. *lunulatum* Hindák | | 159 | 112, 154 | | | |
| *Chlorosarcinopsis* sp. | K4 | | | | | |
| *Coelastrella aeroterrestrica* Tschaikner, Gärtner & Kofler * | | | | | | |
| Strain C_aero K10 | K4 | | | | ITS rRNA, 98.00% identity with authentic strain *Coelastrella aeroterrestrica* SWK1_2, JX513879 | OM522610 |
| Strain 1234 | | | | 153 | ITS rRNA, 99.73% identity with authentic strain *Coelastrella aeroterrestrica* SWK1_2, JX513879 | OM501329 |
| Strain 1236 | | 156 | | | ITS rRNA, 99.87% identity with authentic strain *Coelastrella aeroterrestrica* SWK1_2, JX513879 | OM501330 |

**Table 2.** *Cont.*

| Taxa, Strain | Mutnovsky Volcano | | Gorely Volcano | | Genes, Percentage of Identity with Reference Strain, Accession Number of the Reference Strain | GenBank Accession Number, Publication Information |
|---|---|---|---|---|---|---|
| | 2010 | 2020 | 2010 | 2020 | | |
| Strain 1260 | | 157 | | | ITS rRNA, 100% identity with authentic strain *Coelastrella aeroterrestrica* SWK1_2, JX513879 | OM501331 |
| Strain 1231 | | 158 | | | ITS rRNA, 99.29% identity with authentic strain *Coelastrella aeroterrestrica* SWK1_2, JX513879 | OM501332 |
| *Coelastrella oocystiformis* (J.W.G.Lund) E.Hegewald & N.Hanagata *, strain K1 Coelast1 | K1 | | | | ITS rRNA, 98.49% authentic strain identity with Coelastrella oocystiformis SAG 277-1, JX513887 | OM522609 |
| *Coelastrella terrestris* (Reisigl) Hegewald & N.Hanagata *, strain 1230 | | 158 | | | ITS rRNA, 99.34% identity with *Scotiellopsis terrestris* (Reisigl) Punč. et Kalina SYKOA Ch-045-09, MK480613 | OM501333 |
| *Neocystis mucosa* Krienitz, C.Bock, Nozaki & M.Wolf * | | | | | | |
| Strain K2 N_muc | K2 | | | | 18S rRNA, 99.50% identity with *Neocystis mucosa*, strain SAG 40.88 JQ920367 | OM522658 |
| Strain 1272 | | 159 | | | ITS rRNA, 98.33% identity with *Neocystis mucosa* strain SAG 40.88, JQ920367 | OM501334 |
| Strain K1 N_muc. | K1 | | | | 18S rRNA, 95.97%, identity with *Neocystis mucosa* strain SAG 40.88, JQ920367 | OM522657 |
| Trebouxiophyceae | | | | | | |
| *Chlorella* cf.*chlorelloides* (Naumann) C.Bock. Krienitz & Proeschold *, strain 1261 | | | | 154 | 18S, ITS rRNA, 99.31% identity with *Chlorella chlorelloides* CB 2008/110, HQ111432 | OM501351 |
| *Chlorella* sp.2 | | | K7 | | | |
| *Coccomyxa subellipsoidea* E.Acton * | | | | | | |
| Strain 1249 | | | | 155 | ITS rRNA, 99.80%, identity with *Coccomyxa subellipsoidea* P6065, MH753164 | OM501343 |
| Strain 1235 | | 156 | | | ITS rRNA, 98.39%, identity with *Coccomyxa subellipsoidea* SAG 216-13, HG972978 | OM501344 |
| Strain 1271 | | 159 | | | ITS rRNA, 99.25%, identity with *Coccomyxa subellipsoidea* P6065, MH753164 | OM501345 |
| cf. *Coccomyxa viridis* Chodat | | 159,177 | | | | |
| *Coccomyxa* sp. 1 | K4 | | | | | |
| *Coccomyxa* sp. 2. *, strain 1237 | | | | 112 | 18S, ITS rRNA, 97.19% identity with *Coccomyxa* sp. Obi | OM501346 |
| *Elliptochloris* cf. *reniformis* Darienko & Pröschold *, strain 1291 | | 156 | | | ITS rRNA, 97.73% identity with authentic strain *Elliptochloris reniformis* CAUP H7102, LT560354 | OM501339 |
| *Elliptochloris* cf. *subsphaerica* (Reisigl) Ettl & Gärtner *, strain 1245 | | | | 112 | ITS rRNA, 96.30% identity with authentic strain *Elliptochloris subsphaerica* CAUP H7101, LT560348 | OM501340 |
| *Eremochloris kamchatica* Abdllin&Gontcharov * | | | | | | (Abdullin et al., 2022) |
| Strain 1238 | | | | 154 | ITS rRNA, *Eremochloris kamchatica* Kk5-1 | OM501348 |
| Strain 1246 | | | | 153 | ITS rRNA, *Eremochloris kamchatica* Kk5-1 | OM501347 |
| Strain 1247 | | 156 | | | ITS rRNA, *Eremochloris kamchatica* Kk5-1 | OM501349 |
| *Leptosira obovata* Vischer *, strain K_10-5 | K4 | | | | 18S rRNA, 99.55% identity with authentic strain *Leptosira obovata* SAG 445-1, Z68695 | OM522659 |
| *Lobosphaera incisa* (Reisigl) Karsten et al. * | | | | | | |
| Strain 1248 | | | | 112 | ITS rRNA, 99.76% identity with Lobosphaera incisa chloroplast SAG 2468, LC366923 | OM501338 |

**Table 2.** *Cont.*

| Taxa, Strain | Mutnovsky Volcano | | Gorely Volcano | | Genes, Percentage of Identity with Reference Strain, Accession Number of the Reference Strain | GenBank Accession Number, Publication Information |
|---|---|---|---|---|---|---|
| | 2010 | 2020 | 2010 | 2020 | | |
| Strain 1314 | | | | 101 | ITS rRNA, 99.64% identity with *Lobosphaera incisa* chloroplast SAG 2468, LC366923 | OM501337 |
| *Lobosphaera* sp. *, strain K9 L_inc | K10 | | | | 18S rRNA, 95% identity with Lobosphaera incisa chloroplast CAUP H 4301, LC366922 | |
| *Micractinium* sp. * | | | | 103 | 18S, ITS rRNA, 99.91% identity with *Micractinium* sp. ACSSI 332 | OM501350 |
| *Myrmecia* sp.1 * | | 159 | | | 99.84%, ITS rRNA, *Myrmecia* sp. Ru-s-3-3, MH703746 | OM501355 |
| *Parietochloris pseudoalveolaris* (T.R.Deason & Bold) Shin Watanabe & G.L.Floyd *, strain 1289 | | 157* | + | | ITS rRNA, 99.72% identity with *Ettlia pseudoalveolaris* NV-5, MT735204 | OM501353 |
| *Parietochloris pseudoalveolaris* (T.R.Deason & Bold) Shin Watanabe & G.L.Floyd *, strain 1306 | | 158 | | | ITS, 99.14% identity with *Ettlia pseudoalveolaris* NV-5, MT735204 | OM501354 |
| *Parietochloris* sp. | K1 | | | | | |
| *Pseudococcomyxa* sp. | | | K5 | | | |
| *Stichococcus* sp. 1 | | | | 100 | | |
| *Stichococcus* sp. 2 *, strain 1286 | | | | 102 | ITS rRNA, 88.44% identity with *Stichococcus antarcticus* A.Beck FiSo15/03cVI (M) M-0019691, MH670392 | OM501341 |
| *Stichococcus* sp. 3 *, strain 1270 | | | | 155 | ITS rRNA, 89.86% identity with *Stichococcus antarcticus* A.Beck FiSo15/03cVI (M) M-0019691, MH670392 | OM501342 |
| Ochrophyta | | | | | | |
| *Vischeria magna* (J.B.Petersen) Kryvenda, Rybalka, Wolf & Friedl *, strain K10 V_magna | K4 | | | | ITS rRNA, 97.44% identity with authentic strain *Vischeria magna* SAG 2554, MG596348 | OM522611 |
| *Vischeria* cf. *stellata* (Chodat) Pascher | | 157 | | | | |
| *Vischeria* sp. | | | 99–101 | | | |
| Charophyta | | | | | | |
| *Klebsormidium nitens* (Kützing) Lokhorst *, strain 1290 | | | | 154 | ITS rRNA, 99.14% identity with *Klebsormidium nitens* SAG 335-1a, MN585749 | OM501327 |
| *Klebsormidium* sp1. | K1, K4 | | | | | |
| *Klebsormidium* sp.2. | | 156,158 | | | | |
| *Mesotaenium* sp.* | | 182 | | | | |

Notes: * Strains that were studied using molecular genetic methods; ** Samples from which the taxa or strain was isolated.

## 4. Discussion

It is necessary to note that our study is the first attempt to reveal the algal communities of Mutnovsky and Gorely volcanic soils and their changes at different growth stages by means of morphological and molecular genetic methods. The territory of the Kamchatka Peninsula is practically a "white spot" in terms of the real biodiversity of soil algae and cyanobacteria. Traditional floristic methods do not provide correct species identifications and modern approaches have been used insufficiently in the study of soil algoflora in this area. The use of a polyphasic approach has made it possible to estimate the real biodiversity of terrestrial algae and cyanobacteria on the Mutnovsky and Gorely volcanoes' soils. The importance of using molecular genetic methods in floristics studies has been discussed in numerous publications [74–77].

The use of molecular genetic analysis has allowed us to establish the first findings of a number of taxa in the volcanic soils of Kamchatka. These were *Chlorococcum hypnosporum*, *Chlorococcum lobatum*, *Coccomyxa subellipsoidea*, *Klebsormidium nitens*, *Leptosira obovata*, *Lobosphaera incisa*, *Parietochloris pseudoalveolaris*, *Stenomitos tremulus*, and *Vischeria magna*.

Analysis of species compositions, depending on the height of the sampling, allowed us to describe the changes during the overgrowth of volcanic soils. At an altitude above 1700 m above sea level in the belt of volcanic deposits according to the classification of V. Yu. Neshataeva [39] (samples K6, 102, and 103 of the Gorely volcano), communities of coccoid green algae, such as *Chlorococcum hypnosporum*, *Micractinium* sp., *Pseudococcomyxa* sp., and *Stichococcus* sp. 2, and filamentous cyanobacteria, such as *Microcoleus* cf. *calidus*, have formed on a bare surface without higher plants. These taxa produce a large amount of mucilage, which protects the cells from drying and sticks soil particles together. The abundance of representatives of the Chlorophyta confirms the data in the literature [27], according to which small coccoid green algae play an important role in the colonization of volcanic habitats.

At the next stage of succession in the belt of clumps of herbaceous plants—legumes, sedges, and willows—as a result of the vital activity of photosynthetic microorganisms, the process of accumulation of organic matter of the volcanic substrate began and this was followed by the formation of volcanic destructive soils suitable for the life activity of higher plants, which began to form a mosaic cover [39]. This stage of succession corresponded to samples taken at altitudes from 1053 to 1690 m (samples K3, 156, and 158 of the Mutnovsky volcano, and samples K7-K10, 99–101, 112, and 153 of the Gorely volcano). In these areas, the species richness of algae increases, including green algae, such as *Parietochloris pseudoalveolaris*, *Leptosira obovata*, *Lobosphaera incisa*, *Eremochloris kamchatica*, *Coelastrella aeroterrestrica*, *Chlorosarcinopsis* sp., and others, representatives of Ochrophyta (*Vischeria* sp.), and cyanobacteria (*Stenomitos tremulus*, *Roholtiella* sp., *Stenomitos* sp., *Fischerella* cf. *major*, and *Leptolyngbya* cf. *foveolarum*).

Below the belt of clumps of herbaceous plants, there is a belt of alpine meadows and elfin forests, where the formation of integral phytocenoses was observed. This stage of succession corresponded to samples taken at altitudes below 1065 m above sea level (samples K1, K2, K4, 157,176, and 182 from the Mutnovsky volcano, and sample 154 from the Gorely volcano). A favorable microclimate was created for microorganisms in the rhizosphere of alpine meadows. Such conditions protect them from the winds, ultraviolet radiation, and high temperature. Moreover, algae and cyanobacteria have access to nutrients concentrated in the plant rhizosphere. In addition, in alpine meadows, intense moisture during the warm season due to the melting of snowfields was observed. Under these favorable conditions, a further expansion of the species composition was observed. In this environment, *Coccomyxa* sp. 1, *Neocystis mucosa*, *Parietochloris pseudoalveolaris*, *Parietochloris* sp., *Klebsormidium* sp., *Vischeria magna*, and *Vischeria* cf. *stellata*, were discovered, together with the moisture-loving taxa *Mesotaenium* sp. and *Synechocystis* cf. *salina*.

It should be noted that the boundaries of vegetation belts on volcanoes are conditional and depend on local features. The belts of clumps of herbaceous plants and alpine meadows often overlap. The Mutnovsky and Gorely volcanoes are characterized by low altitudinal vegetation belts, as are other volcanoes of the Southern Volcanic District [39]. This has caused a mosaic pattern in the distribution of algae and cyanobacteria. Some taxa were detected at different altitudes, such as *Eremochloris kamchatica*, *Parietochloris pseudoalveolaris*, *Chlorolobion* cf. *lunulatum*, and *Leptolyngbya* cf. *foveolarum*.

Studies have not shown an unambiguous influence of physical factors, such as temperature, pH, and soil moisture, on the species diversity of algae and cyanobacteria. Perhaps this is due to the specific characteristics of volcanic substrates, which are rocks with small areas of forming soils. In such habitats, micro-habitat features play a crucial role. In those areas where conditions favorable for algae and cyanobacteria obtain (protection from insolation, the presence of vegetation, increased humidity), more intensive growth of these microorganisms is observed.

Despite the absence of general trends, we have found some cases of the influence of environmental factors on algoflora. The resistance of cyanobacteria to high temperatures has been noted in previous studies [44]. In this regard, the finding of *Microcoleus* cf. *calidus*

at a soil temperature of 32 °C is not surprising. As noted above, *Mesotaenium* sp., preferring wet conditions, was found in a wet habitat in an alpine forb meadow.

The descriptions of successions during the colonization of lifeless substrates after volcanic eruptions with the participation of algae and cyanobacteria are given in numerous publications [16–18,27,78,79]. To understand the basic points of these processes in flat areas, long-term observations over many years and even decades are needed. Volcanoes are ideal models for studying soil colonization because from the top of a volcano to its foot it is possible to observe all the stages of the volcano's soils' overgrowth.

A characteristic feature of the species composition of cyanobacteria and algae isolated from the soils of the Mutnovsky and Gorely volcanoes was the dominance of coccoid representatives of the class Trebouxiophyceae. The same feature was also observed in the overgrowth of rocks [80] and industrial dumps [81]. A few cyanobacteria were found in more humid and warmer areas. For example, *Microcoleus* cf. *calidus* was found on the southern slope of the Gorely volcano at the outlet of thermal vapors along the edge of the caldera (sample 103) (Figure 2B), while *Synechocystis* cf. *salina* was isolated from samples of the alpine meadow beyond the pass of the Mutnovsky volcano (sample 159).

In Mutnovsky and Gorely volcanic soils, there are cosmopolitan and widely distributed taxa of algae and cyanobacteria, such as *Bracteacoccus*, *Chlorococcum*, *Coelastrella*, *Coccomyxa*, *Klebsormidium*, *Microcoleus*, *Stichococcus*, and *Vischeria*. F. Rindi et al. [74] called such genera and species "flagship" taxa.

*Bracteacoccus minor* is widely distributed in soils around the world, being a cosmopolitan species [50,51]. It is often found in extreme habitats. This species is widely distributed in spruce forests that are affected by acid rain in the Czech Republic [82]. The algae are one of the most common species on brown coal dumps in Central Europe [83]. The species is also found in desert crusts in North America and in halophilic crust algal communities in potash tailing piles [77]. *Bracteacoccus minor* was detected in wet rocks in the territory of the Great Smoky Mountains [84] and illuminated the entrance zones of the Njegoš Pećina cave in Montenegro [85].

*Chlorococcum* is one of the most common genera in terrestrial ecosystems [50,51]. *Chlorococcum hypnosporum* was detected in forest steppes, steppes in Ukraine, in the Ukrainian Carpathians, and in the Crimean Mountains [86]. This taxon was also found among the airborne algae, detected at the Westinghouse Environmental Station Laboratory in Raleigh, North Carolina [87]. *Chlorococcum hypnosporum* was identified on Kryvyi Rih iron ore tailing dumps (Ukraine) [88]. *Chlorococcum lobatum* is a member of soil algal communities on the active volcano of Deception Island in Antarctica [20]. It was also detected in volcanic soils distributed in different zones of Ukraine [86], as well as in communities of mosses collected from rocks and soils in the territory of the Sokhondo biosphere nature reserve (Zabaykalsky region, Russia) [89].

*Vischeria magna* (as *Eustigmatos magnus*) has a wide distribution area around the world, being found in Africa [90], the USA [91–93], Ukraine [86], Azerbaijan [94], Kazakhstan [95], the Czech Republic [82,83], Germany [83], and Russia [96,97]. This species is found in areas with extreme habitat conditions, e.g., in different types of solonetzes. It is found in crustal solonetzes in the chernozem, chestnut and brown zones, in medium solonetzes–chernozem and chestnut zones, in deep solonetzes–chernozem, chestnut, brown zones, in solonetzes solonetzes–chernozem zones, and in meadow solonetzes–black earth zones [95]. In addition, *Vischeria magna* has been found in the acid rain zones of spruce forests in the Czech Republic and Germany [83].

*Coelastrella aeroterrestrica* was described from soils in the Alps [70]. Representatives of the Scenedesmaceae are cosmopolitan and distributed in different regions of the world [98]. Thus, this species was found in microbiotic crusts of the Arctic Spitsbergen [99], in soil samples from the Pirin Mountains in Bulgaria [71], and in a reddish crust on a piece of foam plastic in the area of the White Sea Biological station of Moscow State University in the Kandalaksha Bay of the White Sea (Lukhinsky district, Republic of Karelia, Russia) [100]. A. Yu. Nikulin et al. [101] noted the high occurrence of *Coelastrella aeroterrestrica* in soil

samples under vegetation with *Sasa kurilensis* (Rupr.) Makino and Shibata on the Iturup Island (Kuril Islands).

In the first description of *Coccomyxa subellipsoidea*, it was indicated that the species is widely distributed in all parts of the British Isles—on wet rocks and stones, and in cold greenhouses in the form of a green slimy coating on both glass and wooden elements, given sufficient moisture [102]. *Coccomyxa subellipsoidea* was isolated from dry algal peat in Victoria Land in Antarctica [103] and the entire genome sequence was determined for this organism; later, it was described as a model organism of adaptation to cold [103,104]. The studied species is also known as a lichen phycobiont, *Lichenomphalia umbelifera* (L.) Redhead, Lutzoni, Moncalvo & Vilgalys (found in Innsbruck, Austria) [105] and *Lichenomphalia meridionalis* (Contu and La Rocca) P.A. Moreau and Courtec. that was obtained from roadside andosols at an altitude of 1200 to 1900 m in Japan [106].

Representatives of the genus *Stichococcus* are very common and can be found in almost all types of habitats, including fresh water, brackish water, marine environments, hot acid springs, and snow [50,55,107,108]. Representatives of this genus were recognized as key organisms in the successions of the formation of biological soil crusts [109]. *Stichococcus* has also been found in the harsh terrestrial and freshwater environments of the Arctic and Antarctica [110–112].

*Microcoleus* is one of the most common cyanobacteria in terrestrial ecosystems [113–115]. *Microcoleus calidus* is characterized by its rare distribution in comparison with other species of *Microcoleus*. It was found in freshwater ecosystems in Nueve Leon State (Mexico) [116]. Possibly, this taxon is more common in different habitats.

*Klebsormidium* is a cosmopolitan genus, distributed in terrestrial habitats around the world [117–120]. *Klebsormidium nitens* was found at pH 4.3 and in metal contaminants on the former mining site of Schwarzwand (Salzburg, Austria) [121]. It was also detected in alpine biological soil crusts in Austria and Italy [120].

In the study of samples from the Avachinsky, Tolbachinsky, and Shiveluch volcanoes, nineteen species of cyanobacteria and algae were found [122]. Cosmopolitan species, which are characterized by wide distribution and resistance to extreme habitat conditions, were identified, including the cyanobacteria *Nostoc* cf. *punctiforme* Hariot and *Trichocoleus* cf. *hospitus* (Hansgirg ex Gomont) Anagnostidis and the algae *Bracteacoccus minor*, *Pseudococcomyxa simplex*, and *Klebsormidium flaccidum*, species of the genus *Chlorella*. In a study of the soil algae and cyanobacteria on an active Antarctic volcano on Deception Island, species compositions very similar to those observed on the Mutnovsky and Gorely volcanoes, including widely distributed and cosmopolitan taxa, were described [20]. Soil algal communities included genera *Leptolyngbya*, *Phormidium*, *Chlorella*, *Chlorococcum*, *Pseudococcomyxa*, and *Stichococcus*. Possibly, cosmopolitan taxa have unique resistance mechanisms that allow them to survive in extreme conditions.

Thus, the investigation of the biodiversity of algae and cyanobacteria of soils of the Mutnovsky and Gorely volcanoes using a polyphasic approach allowed us to conduct precise taxa identification at least at the genus level. In the species compositions of the volcanic soils of the studied area, representatives of cosmopolitan genera prevailed. Analysis of the algal communities at different altitudes revealed an expansion in species richness with increasing distance from the tops of the volcanoes. This study presents new knowledge about the role of algae and cyanobacteria in the overgrowth of lifeless volcanic substrates, as well as their biogeography and ecology.

**Author Contributions:** Conceptualization, L.A.G. and R.Z.A.; methodology, L.A.G.; software, A.Y.N. and V.Y.N.; validation, L.A.G., A.Y.N. and V.Y.N.; formal analysis, R.Z.A., L.A.G. and V.B.B.; investigation, R.Z.A., A.Y.N., V.Y.N., V.B.B. and L.A.G.; resources, L.A.G.; data curation, R.Z.A. and L.A.G.; writing—original draft preparation, R.Z.A. and L.A.G.; writing—review and editing, L.A.G. and A.Y.N.; visualization, L.A.G.; supervision, L.A.G. and A.Y.N.; project administration, L.A.G.; funding acquisition, L.A.G. All authors have read and agreed to the published version of the manuscript.

**Funding:** The reported study was funded by the Russian Foundation for Basic Research, project number 20-04-00814 a. The research was carried out within the state assignment of the Ministry of Science and Higher Education of the Russian Federation (theme no. 121031000117-9).

**Institutional Review Board Statement:** Not applicable.

**Data Availability Statement:** Not applicable.

**Acknowledgments:** The authors are thankful to A. A. Gontcharov and Sh. R. Abdullin (Federal Scientific Center of the East Asia Terrestrial Biodiversity) for samples and valuable discussions, and Yu. Z. Gabidullin for help with preparation of the figure plates.

**Conflicts of Interest:** The authors declare no conflict of interest.

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
