# Peer review of "Study of Biodiversity of Algae and Cyanobacteria of Mutnovsky and Gorely Volcanoes Soils (Kamchatka Peninsula) Using a Polyphasic Approach"

_diversity, doi:10.3390/d14050375_

Round 1

Reviewer 1 Report

The manuscript is interesting but contains many methodological errors that need to be corrected Figure 1. It should be cited on the Study site.

Figure 1. Where do the photos come from? Did the authors take them themselves? Please provide the source,

Line 145. I do not understand why the authors write, "Samples of soil were collected in August 2010 and August 2020 (Table 1) (Figures 1, 2). In August 2010 thirteen samples were taken, and diversity of diatom algae from this samples was studied [33] (Fazlutdinova et al., 2021). " How does this relate to the presented work?

Line 147. "In present strudy only tren samples (K1-K10) were used." I do not understand this sentence; please explain Table 1. in the table, the authors present the test results, in the table, there is information in the nember row. 4) 50-58% moisture has Rocks? It is unlikely to happen. Same situation on March 13 and 14 - please explain it.

 Line 151. When did the authors take the photos? Humidity% for soil is used "moisture". The authors describe in detail how they took the samples for testing, but there is no information on how soil samples were taken for pH determination and how it was determined in KCl or in H2O. How were soil samples taken for Moisture determination? The table shows the results. Where did they come from? From the samples taken into paper bags and air-dried, the moisture of the soil cannot be determined! - in accordance with the requirements of the journal, I apply 5.55, etc. And in the table we find 5.55; please correct it.

 Line 220. the names of species and genera are written in italic

Line 591. References, please prepare the year of publication in Bold in accordance with the journal's requirements. The name of the magazine is Italic.

 Line. 592; 594; 596 etc.

Author Response

Dear reviewer,

We are very grateful to you for your valuable comments and advice, which helped us a lot to improve the quality of the MS. We did the major revision of the MS. The primary new additions and corrections were: 1) the correction of methodological errors; 2) phylogenetic tree of Scenedesmaceae for better understanding the positions of Coelastrella strains; 3) the information about soil temperature; 4) the explanation about influence of physical factors; 5) the information about the number of taxa in different sites; 6) using MDPI English correction service; 7) corrections the errors in MS and references list.

We have tried to fix all the errors that you have pointed out.

The manuscript is interesting but contains many methodological errors that need to be corrected Figure 1. It should be cited on the Study site.

Yes, we agree. The errors were corrected. Figure 1 was cited on the Study site.

Figure 1. Where do the photos come from? Did the authors take them themselves? Please provide the source.

Thank you very much for the comment! In Figure 1 we used the map, which was created for our publication in Microorganisms MDPI https://doi.org/10.3390/microorganisms9091851. We modified it, added the new sampling sites and corrected  contours of volcanoes using GIM software. We wrote the source in the legend to the figure.

Line 145. I do not understand why the authors write, "Samples of soil were collected in August 2010 and August 2020 (Table 1) (Figures 1, 2). In August 2010 thirteen samples were taken, and diversity of diatom algae from this samples was studied [33] (Fazlutdinova et al., 2021). " How does this relate to the presented work?

Yes, we agree. The errors were corrected.

Line 147. "In present strudy only tren samples (K1-K10) were used." I do not understand this sentence; please explain Table 1. in the table, the authors present the test results, in the table, there is information in the nember row. 4) 50-58% moisture has Rocks? It is unlikely to happen. Same situation on March 13 and 14 - please explain it.

We delete this unclear sentence. We delete the data about pH and soil moisture in case of samples, which were taken from rocks.

Line 151. When did the authors take the photos? Humidity% for soil is used "moisture". The authors describe in detail how they took the samples for testing, but there is no information on how soil samples were taken for pH determination and how it was determined in KCl or in H2O. How were soil samples taken for Moisture determination? The table shows the results. Where did they come from? From the samples taken into paper bags and air-dried, the moisture of the soil cannot be determined! - in accordance with the requirements of the journal, I apply 5.55, etc. And in the table we find 5.55; please correct it.

In Figure 2 we used our own pictures, which were taken during the sampling. We corrected the “humidity” to “soil moisture”. We didn’t measure pH and humidity during our study. The data about this factors we took from the literature. We wrote this information I notes for Table 1. We delete the data about pH and soil moisture in case of samples, which were taken from rocks. In the pH designations, commas have been corrected to dots.

 Line 220. the names of species and genera are written in italic

The errors were corrected.

Line 591. References, please prepare the year of publication in Bold in accordance with the journal's requirements. The name of the magazine is Italic.

The errors were corrected.

 Line. 592; 594; 596 etc.

The errors were corrected.

Sincerely yours,

                     Rezeda Allaguvatova and Lira Gaysina

Reviewer 2 Report

The paper reveals new data on the diversity of cyanobacteria and microalgae at two volcanoes on the Kamchatka Peninsula.
This is a paper with some detail on the biology and diversity of these organisms, collected from different soils/altitudes of the volcanoes, and therefore presents a high diversity of organisms. The methods used are adequate, having resorted to morphological and genetic identification. 
Numerous taxa are listed, some new to the site. 
In the results, a relation is made between this diversity and the type of substratum where these organisms are found. The paper is consistent, relating the results to those obtained by other authors for similar ecosystems. 
I would have liked to see a correlation between the physical factors (substrate, pH and temperature) that were collected and the diversity found. 
Still, I believe the paper is interesting enough to be published.
Some errors need to be corrected and are marked in the attached document. 

Author Response

Dear reviewer,

We are very grateful for the valuable comments that helped improve the quality of the manuscript.

We are very grateful to you for your valuable comments and advice, which helped us a lot to improve the quality of the MS. We did the major revision of the MS. The primary new additions and corrections were: 1) the correction of methodological errors; 2) phylogenetic tree of Scenedesmaceae for better understanding the positions of Coelastrella strains; 3) the information about soil temperature; 4) the explanation about influence of physical factors; 5) the information about the number of taxa in different sites; 6) using MDPI English correction service; 7) corrections the errors in MS and references list.

We corrected the MS according your suggestions:

I would have liked to see a correlation between the physical factors (substrate, pH and temperature) that were collected and the diversity found. 

We tried to find the correlations between the diversity of algae and cyanobacteria. Unfortunately, we were not able to find strong correlations, only isolated cases of influence. We wrote the explanation in Discussion. But your suggestion was very important, it helped us to discuss our results from a new perspective.

Some errors need to be corrected and are marked in the attached document. 

Special thanks for your edits in the pdf file. We corrected all errors and wrote the soil temperatures in the Table 1.

Sincerely yours,

                     Rezeda Allaguvatova and Lira Gaysina

Round 2

Reviewer 1 Report

The authors took into account most of the comments, but there are still some bugs
  Why did the authors include figure 4 if it is not cited in the text?
Shoulder citation Rycina 4. correct
References are not prepared in accordance with the journal's requirements
  Figure 4 is illegible - correct
  the article is prepared chaotically, please correct Table 2 so that the discoverer is correctly assigned under the species name.

Author Response

Dear reviewer,

We are very grateful to you for a thorough reading of our article and valuable comments. We fixed the bugs by highlighting the corrections by tracking and blue color.

The authors took into account most of the comments, but there are still some bugs

 Why did the authors include figure 4 if it is not cited in the text?

Figure 4 was cited in the text on lines 1194 and 1196. 

Shoulder citation Rycina 4. Correct

We have corrected Figure 4.

References are not prepared in accordance with the journal's requirements

 Figure 4 is illegible – correct

We have made the tree in Figure 4 more legible. We attach .pdf and .eps files of this image and ask you to use .pdf in the final layout to maintain the quality.

  the article is prepared chaotically, please correct Table 2 so that the discoverer is correctly assigned under the species name.

Table 2 was corrected according to your advice.

Sincerely yours,

                      Lira Gaysina
